# Predicting the burden of acute malnutrition in drought-prone regions of Kenya: A statistical analysis

**Francesco Checchi**[1]*, **Rahaf AbuKoura**[1], **Suneetha Kadiyala**[2], **Mara Nyawo**[3], **Lucy Maina**[4]

**1** Department of Infectious Disease Epidemiology and International Health, Faculty of Epidemiology and Population Health, London School of Hygiene and Tropical Medicine, London, United Kingdom, **2** Department of Population Health, Faculty of Epidemiology and Population Health, London School of Hygiene and Tropical Medicine, London, United Kingdom, **3** United Nations Children's Fund, East and Southern Africa Regional Office, Nairobi, Kenya, **4** United Nations Children's Fund, Kenya Country Office, Nairobi, Kenya

* Francesco.checchi@lshtm.ac.uk

## Abstract

In drought-prone regions, timely and granular predictions of the burden of acute malnutrition could support decision-making. We explored whether routinely collected and/or publicly available data could be used to predict the prevalence of global and severe acute malnutrition, as well as the mean weight-for-height Z-score and middle-upper-arm circumference for age Z-score, in arid- and semi-arid regions of Kenya, where drought is projected to increase in frequency and intensity. The study covered six counties of northern Kenya and the period 2015–2019, during which a major drought occurred. To validate models, we sourced and curated so-called SMART anthropometric surveys covering one or more sub-counties for a total of 79 explicit survey strata and 44,218 individual child observations. We associated these surveys' predictors specified at the sub-county or county level, and comprising climate food security, observed malnutrition, epidemic disease incidence, health service utilisation and other social conditions. We explored both generalised linear or additive models and random forests and quantified their out-of-sample performance using cross-validation. In most counties, survey-estimated nutritional indicators were worst during the October 2016-December 2019 drought period; the drought also saw peaks in insecurity and steep vaccination declines. Candidate models had moderate performance, with random forests slightly outperforming generalised linear models. The most promising performance was observed for global acute malnutrition prevalence. The study did not identify a model that could very accurately predict malnutrition burden, but analyses relying on larger datasets with a wider range of predictors and encompassing multiple drought periods may yield sufficient performance and are warranted given the potential utility and efficiency of predictive models in lieu of assumptions or expensive and untimely ground data collection.

**Data availability statement:** Data and analysis code are available on https://github.com/francescochecchi/ken_malnut_prediction/.

**Funding:** This study was funded by UNICEF's East and Southern Africa Regional Office (ref. ESARO/SSFA/2019-001; funding recipient: FC), and by the Innovative Methods and Metrics for Agriculture and Nutrition Actions (IMMANA) programme (funding recipient: SK), which is funded by the Gates Foundation and the United Kingdom Foreign, Commonwealth and Development Office (FCDO). The Gates Foundation and FCDO had no role in study design, data collection and analysis, decision to publish, or preparation of the manuscript. As UNICEF commissioned this research to address a programmatic question, MN and LM, who work for UNICEF, took part in data collection, the decision to publish and the preparation of the manuscript.

**Competing interests:** The authors have declared that no competing interests exist.

## Background

Acute malnutrition or wasting is a leading underlying factor behind childhood mortality and untoward outcomes of pregnancy [1]. Over the past few decades, extremely high prevalences of acute malnutrition have been observed in crisis-affected populations, particularly during periods of acute food insecurity [2–5]. Drought conditions pose a major threat to food security within arid and semi-arid regions of the Horn of Africa, including northern Kenya, and are projected to occur with greater frequency and intensity in this region due to climate change [6].

Knowing the population burden (prevalence) of acute malnutrition serves multiple purposes, including the selection of appropriately scaled food security, nutritional and health interventions (e.g., cash transfers; blanket feeding distributions; management of severe and/or moderate malnutrition), resource mobilisation, logistical planning for an expected incident level of cases and monitoring of an ongoing response [7]. In drought-affected settings, the mainstay of measuring acute malnutrition prevalence is so-called Standardised Monitoring and Assessment of Relief and Transitions (SMART) surveys, highly standardised data collection exercises usually conducted at the administrative level 2 scale or below and targeting children aged 6–59 months old (mo), the main at-risk group [7].

Over the past decade, SMART surveys have benefited from increased technical and software support [8], with evidence of improved quality [9]. However, they remain somewhat costly to implement, with global estimates ranging from 10-40,000 USD per survey depending on transport, staffing, and other factors [10] and cannot feasibly be conducted on an ongoing basis across all potentially affected areas. While sentinel-based approaches have also been attempted [11,12] and reductions to the sample size of surveys have been shown to be possible [13], no method is currently available that can efficiently provide estimates with timeliness and geographic granularity.

Across different areas of public health, statistical approaches underpinned by analysis of multiple existing or routinely collected data sources have been used to provide estimates based solely on secondary, desk-based analysis [14–17]. Such an approach could offer a complementary (and much cheaper) method to surveys and support governments and humanitarian actors to detect worsening conditions and respond efficiently and on time to these. We previously showed that predictive small-area estimation models did not accurately predict acute malnutrition prevalence in South Sudan and Somalia [18]. Here, we report a similar study focussed on the arid- and semi-arid counties of Kenya and relying on a somewhat different range of data.

## Methods

### Ethics statement

This study involves secondary analysis of de-identified data. Ethical approval was obtained through the Amref Ethics and Scientific Review Committee, the competent committee in Kenya (ref. ESRC P723/2019) and the London School of Hygiene and Tropical Medicine Ethics Committee (ref. 15334, amendment 3).

## Study population and period

We included in the analysis the counties of Baringo, Garissa, Isiolo, Mandera, Marsabit, Samburu, Tana River, Turkana, Wajir and West Pokot, and all sub-counties within these (Fig 1; in Kenya, sub-counties have limited administrative function and have seen considerably boundary changes: we relied on a list and boundary provided by the United Nations in 2019 [19]). The period of analysis was from January 2015 to December 2019 inclusive, which included a period of acute drought across the Horn of Africa unfolding between October 2016 and the end of 2018. The region of analysis had an estimated population of 8.17 million (see Data Sources and management) at the midpoint of the period.

## Data sources and management

**Anthropometric data.** We obtained all available raw datasets and reports of SMART anthropometric surveys carried out within the study area and timeframe. All surveys relied on a multi-stage cluster sampling design, and some had explicit strata corresponding to either single sub-counties or groupings of these. Each survey sought to include any children aged 6–59 months old (mo) within households sampled. Datasets had various formats; we sought within each the following minimal variable set: cluster number, child ID, age in months or days, sex, weight to the nearest 0.1 Kg, height in cm, presence of bilateral oedema and middle-upper arm circumference (MUAC) in cm or mm. To the extent possible, we identified the sub county location of individual survey clusters using household coordinates where these were included in the dataset or names of localities where each cluster fell if contained in the report or the data. We computed anthropometric indices of interest (weight-for-height Z-score, WHZ; MUAC-for-age Z-score, MUACZ) based on WHO 2006 growth chart standards using the R `anthro` package [20]. We then applied the following exclusion criteria to each child observation: (i) minimal set of variables (see above) incomplete; (ii) age outside the 6–59mo range; (iii) sub-county location of cluster unknown; (iv) WHZ and/or MUACZ <> 5SD, indicating implausible values as per WHO recommendations [21]. We defined global (GAM) and severe (SAM) acute malnutrition as WHZ<-2SD and/or bilateral oedema, and WHZ<-3SD and/or bilateral oedema, respectively.

**Predictors of malnutrition.** We wished to explore statistical models that could accurately predict anthropometry, and which government and humanitarian actors could apply routinely without needing to collect ad hoc ground data. We therefore searched for public or non-public and routinely collected/compiled data that would directly or by proxy represent any of a range of factors that would plausibly causally affect or at least correlate with acute malnutrition. Datasets also needed to feature sufficient geographic granularity (at least by county, preferably by sub-county) and consistently cover the geography and time period analysed. Guided by two previously published causal frameworks [18,22] of acute malnutrition (Fig A and Fig B in S1 Appendix), we worked with the Ministry of Health of Kenya and UN agencies to identify suitable datasets, while also searching for public sources online.

While proximate indicators of acute malnutrition, such as household food insecurity, care practices, and household environment, are well-recognised within the conceptual frameworks, these are typically available only through surveys like DHS and MICS, which are infrequent and reported at higher administrative levels. As these could not be linked with SMART survey clusters at the sub-county level without strong assumptions, we excluded them to preserve spatial and temporal compatibility across predictors. We therefore focused on indicators that were available at the sub-county (or county) level for all or most of the 2016–2023 study period.

Where sub-county indicators from administrative sources (e.g., DHIS2) were considered but found to be unavailable, inconsistently reported, or inaccessible, such as skilled birth attendance or facility delivery, we included publicly available modelled estimates (e.g., WorldPop) that provided full spatial coverage and alignment with the study period. These variables were not intended as precise causal measures, but as pragmatic proxies for underlying constructs such as access to maternal health services. While we recognise their limitations, they allowed us to maintain comparability across strata and enable operational use in prospective applications. Table 1 lists predictors that, after lengthy searches and exploratory

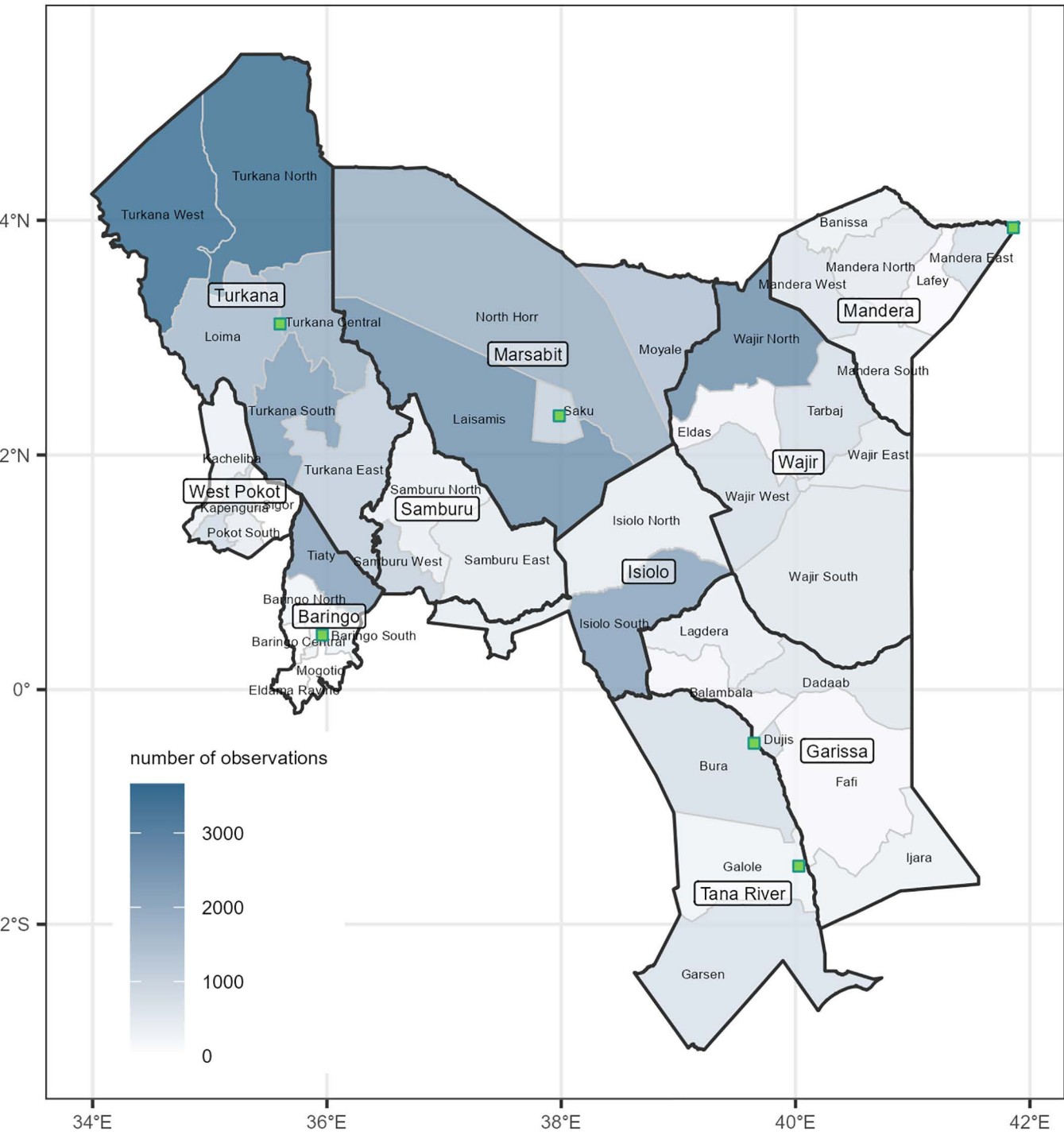

**Fig 1. Map of the sub-counties and counties included in the study.** The extent of shading indicates the number of survey observations available for this analysis. Green squares indicate the location of sentinel markets. Base layer source: United Nations Office for Coordination of Humanitarian Affairs, Humanitarian Data Exchange (https://data.humdata.org/dataset/cod-ab-ken). License: Creative Commons Attribution for Intergovernmental Organisations (CC BY-IGO) (https://data.humdata.org/faqs/licenses).

**Table 1. List of predictors included in the analysis, and their key characteristics.**

| Predictor and variable | Scale | Source (URL) | Notes |
|---|---|---|---|
| Drought conditions | | | |
| Rainfall abnormality | Sub-county | Climate Hazards Centre, Climate Hazards Group Infrared Precipitation with Stations (CHIRPS), version 2.0 [23] (https://data.chc.ucsb.edu/products/CHIRPS-2.0/africa_monthly/tifs/) | Sourced monthly precipitation totals with 0.05° resolution, and converted these to a standardized precipitation index (SPI), which expresses deviance from the historical average, using the r SPEI package [24]. |
| Vegetation index | Sub-county | Intergovernmental Authority on Development Climate Prediction and Applications Centre, Global Monitoring for Environment and Security and Africa Support Programme (https://gmes.icpac.net/data-center/vgt-ndvi) | Sourced monthly normalized difference vegetation index (NDVI) and standardized NDVI (i.e., relative to the historical average) with 1Km resolution. |
| Food security | | | |
| Price of food staples | Sub-county | United Nations World Food Programme (https://data.humdata.org/dataset/wfp-food-prices-for-kenya) | Only the price of white maize was available consistently over the time period analysed. Monthly time series from the six available sentinel markets (Garissa; Hola, Tana River; Lodwar, Turkana; Mandera; Marigat, Baringo; Marsabit; see Fig 1) were adjusted for inflation to compute 2015 Kenyan shillings (KES) reference prices using the agriculture / forestry sector deflator annual indices provided by the UN Food and Agriculture Organisation (https://data.humdata.org/dataset/faostat-food-prices-for-kenya). Each time series was moderately smoothed to attenuate outliers, and attributed to the sub-county nearest to the market as defined by distance to each sub-county's centroid. |
| Observed incidence of acute malnutrition | | | |
| Incidence of acute malnutrition admissions to treatment services | Sub-county | United Nations Children's Fund (UNICEF) (non-public) | Stratified into severe and moderate (-3SD < WHZ < -2SD with no bilateral oedema) acute malnutrition cases, and inclusive of monthly outpatient, inpatient and repeat admissions. Not available for West Pokot county. |
| Occurrence of epidemic disease | | | |
| Incidence of cholera and measles | County | Ministry of Health (non-public) | Monthly cases, suspected or confirmed. |
| Utilisation of health services | | | |
| Vaccination coverage | County | Ministry of Health (non-public) | Monthly number of doses of different routine childhood vaccines administered, by dose in the recommended scheduled. Used pentavalent (diphtheria-pertussis-tetanus-*Haemophilus influenzae* type B-hepatitis B) dose 3 and measles-mumps-rubella dose 1 as the signal doses, divided by population to express a correlate indicator of coverage. |
| Proportion of the population with access to safe assisted births (static) | Sub-county | WorldPop [25] (https://hub.worldpop.org/geodata/summary?id=1263) | Probabilistic prediction based on re-analysis of household survey data, with resolution approximately 300 m² centered in the year 2015. Included due to full sub-county coverage and availability during the study period. Used as a proxy for access to maternal health services in the absence of reliable, complete administrative data. |
| Wider social conditions | | | |
| Literacy | Sub-county | WorldPop [26] (https://hub.worldpop.org/geodata/summary?id=1261) | Probabilistic prediction of the proportion of women aged 15–49yo classed as literate, as of 2008–2009. Resolution approximately 1Km². |
| Schooling attendance | County | Kenya National Bureau of Statistics [27] (https://data.humdata.org/dataset/kenya-distribution-of-population-age-groups-3-years-and-above-by-school-attendance-status-special) | Proportion of children attending primary education as of March 2020, based on administrative records. |
| Insecurity | Sub-county | Armed Conflict Location and Event Data Project [28] (https://data.humdata.org/dataset/kenya-acled-conflict-data) | Georeferenced instances of insecurity events and fatalities as abstracted from media and civil society sources. All categories of events were retained when aggregating data to monthly and subcounty level. |

data cleaning, met all the above criteria and were used to develop candidate models: the table also provides specific data management steps for each. For all time-varying predictors, we computed right-aligned rolling means of values over the previous 3 and 6 months, which also resolved minor missingness instances. We categorised heavily skewed predictors into quintiles, with a zero category if the latter was frequent and epidemiologically meaningful, e.g., in the case of cholera incidence. Summary statistics for each predictor variable are shown in Table C in S1 Appendix.

**Population denominators.** So as to make the value of all predictors comparable across time and sub-county, we divided acute malnutrition admissions, cases of epidemic disease, vaccine doses administered and incidence of insecurity events by population. As sub-counties are not presented as a unit in existing census projections, we used annual WorldPop estimates [29,30], constrained to match census-based demographic projections but available at 100m$^2$ pixels (https://data.worldpop.org/GIS/Population/), which we aggregated to sub-county level and interpolated to provide a monthly value.

## Statistical analysis

To adequately propagate survey non-systematic error, we fitted models to individual child observations, with the concurrent value of predictors specified either at the sub-county (if possible) or county level. We developed models for four outcomes: GAM, SAM (binary, assumed to be binomially distributed), WHZ and MUACZ (continuous, assumed to be Gaussian-distributed). We explored two classes of models: (i) generalised linear or additive models featuring combinations of predictors, with or without additive smoothing terms and/or a random effect for county to capture additional variability, using the R `mgcv` package [31]; and (ii) random forest classification using the `ranger` package [32], using 1000 trees and maximum five variables to split at within each tree node.

We first fitted univariate generalised linear or additive models of each predictor and each outcome, monitoring the resulting Akaike Information Criterion (AIC). We also selected between the two alternative rolling mean periods (3 versus 6 months) based on lowest AIC in univariate analysis (Fig E in S1 Appendix). To select the most performant generalised linear/ additive model, we entered variables into models from lowest (best) to highest univariate AIC, with or without a smoothing term for continuous time-varying variables and avoiding pairs of highly correlated variables (Fig D in S1 Appendix). Faced with a large set of potential predictor combinations, including varying lags, initial AIC-based screening of the most promising predictors offers a reasonable approach to identify a smaller set of highly performant models. We also tested mixed models with county as the random effect, but these tended to predict less well out-of-sample than fixed-effects models.

As models grew in complexity, we tracked their performance for out-of-sample prediction (the key metric by which to evaluate their likely utility) using leave-one-out cross-validation (LOOCV), with folds specified at the individual survey explicit stratum level. LOOCV is a simple technique to test the model's performance on data it has not been trained on: briefly, the model is trained on all survey strata within the dataset bar one (the latter is known as the 'fold'), and used to predict observations in the fold; this process is repeated, cycling through the dataset until all predictions on all possible folds are made: the mean prediction error across the folds is expected to better represent the model's out-of-sample validity than its accuracy on training data only.

As metrics of performance, we computed models' absolute bias relative to predictions, mean absolute error, probability of predicting the outcome within given thresholds precision bounds, and sensitivity against alternative crisis severity thresholds of 10% and 15% (i.e., the proportion of observations above the threshold that were also classified as above the threshold by the model): while absolute thresholds are discouraged in nutritional surveillance [7], in practice these cut-offs are used by Integrated Phase Classification analyses to help distinguish phases 3 (crisis) and 4 (emergency) [33]. Lastly, we implemented the following diagnostics to verify key assumptions of generalised models: (i) for logistic models of SAM and GAM, we graphed leverage due to influential values, checked that each continuous predictor had a reasonably linear association with the predicted logit and tested for multicollinearity; (ii) for additive Gaussian models of WHZ and MUACZ, we checked for normality in the distribution of residuals, heteroskedasticity and deviance residuals against approximate

theoretical quantiles of the deviance residual distribution. These diagnostics are shown in Fig F, Fig G, Fig H, Fig I, Fig J, Fig K, Table D and Table E in S1 Appendix). All analysis was done in R Statistical Software [34]. Data and analysis code are available on https://github.com/francescochecchi/ken_malnut_prediction/.

## Results

### Anthropometric survey patterns

Altogether, the analysis relied on data from 79 explicit survey strata, comprising 44,536 child observations, of which 44,218 (99.3%) were eligible for analysis after exclusion criteria were applied, or a mean of 560 observations per stratum. Individual survey-stratum attrition and estimates are reported in Table A and Table B in S1 Appendix, respectively. Survey coverage was uneven across the region studied, with sub-counties in Turkana accounting for most observations and Garissa the fewest (Fig 1).

Survey-stratum point estimates are shown in Fig 2 for SAM and GAM and Fig 3 for WHZ and MUACZ, with lines connecting sequential stratum estimates. Generally, the highest SAM/GAM prevalences (lowest mean WHZ/MUACZ) were observed during 2017, but there were considerable differences among strata within the same county (Marsabit, Turkana) and in some counties (Samburu, Wajir) there was comparatively less variation.

### Predictor patterns

County-level trends in each predictor are shown in Fig 4. The onset of drought appeared more visible when tracking NDVI and standardised NDVI than rainfall abnormalities. Cereal prices increased in all sentinel markets (Fig C in S1 Appendix) and continued increasing in Garissa and Mandera. During the drought period, MAM and SAM admissions increased markedly in Turkana and Marsabit, with a more delayed peak in Isiolo. The drought period was also generally concurrent with peaks in insecurity events and deaths, and with a steep fall in vaccination output. The period also saw epidemics of cholera and measles, but these occurred mostly outside the drought period, with the largest peak in Tana River.

### Predictive model performance

**Generalised linear or additive models.** The goodness-of-fit (AIC) of each predictor variable on univariate analysis is summarised in Fig E in S1 Appendix. By this metric, (S)NDVI and recent incidence of SAM or MAM treatment admissions had the strongest associations with each outcome. Fig 5 shows predictions versus observations on cross-validation for the highest-performance multivariate models identified for each outcome.

Generalised linear models were retained for GAM and SAM prevalence, but additive versions of these were marginally superior for mean WHZ and MUACZ (Table 2). All models had negligible bias, but only a minority of predictions fell within the stricter of the two error bounds suggested for each outcome. Models had reasonably high sensitivity for predicting instances of high GAM and SAM prevalence at thresholds of 15% and 2% respectively, but low sensitivity when these thresholds were raised to 20% and 5%.

**Random forest models.** Prediction-versus-observations graphs for random forest models of each outcome are shown in Fig 6. Random forest performance with up to three variables to split data by at each node appeared visually marginally better than that of generalised linear/additive models for all outcomes except MUACZ (Table 3), though the comparison of sensitivity between the two model classes is based on few observations, at least for the more severe of the two thresholds evaluated. Random forests with up to five variables to split data by per node also achieved moderate performance (Fig L in S1 Appendix).

## Discussion

### Key findings

While nutritional surveillance in Kenya has become more systematic in recent years [35], a dearth of ground data continues to hamper the timeliness and geographic targeting of nutritional and other public health responses to

PLOS Global Public Health

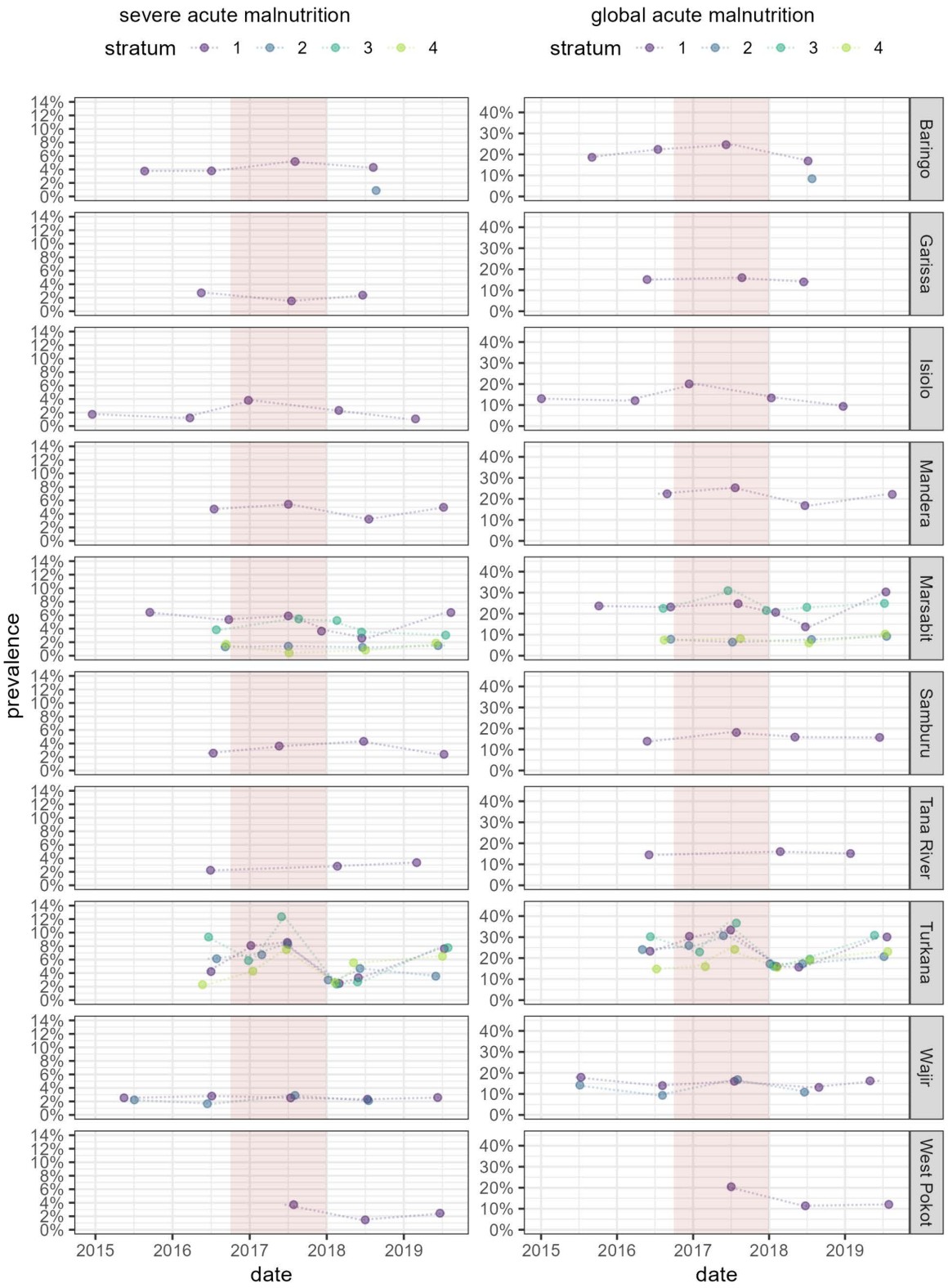

**Fig 2. Point estimates of severe and global acute malnutrition prevalence, by county, stratum and survey date.** The pink-shaded band indicates the drought period (October 2016 to December 2018).

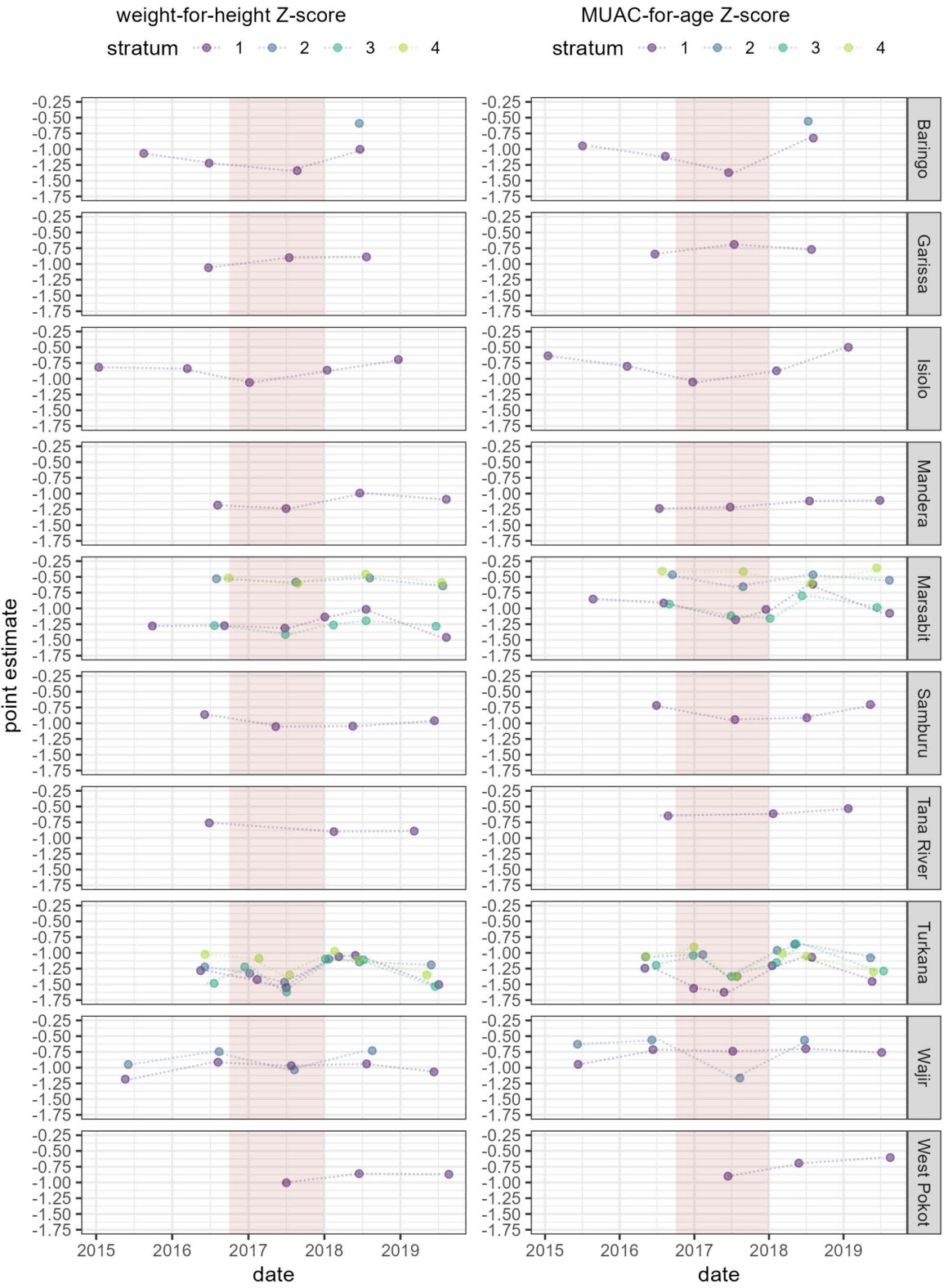

**Fig 3. Point estimates of weight-for-height Z-score and MUAC-for-age Z-score, by county, stratum and survey date.** The pink-shaded band indicates the drought period (October 2016 to December 2018).

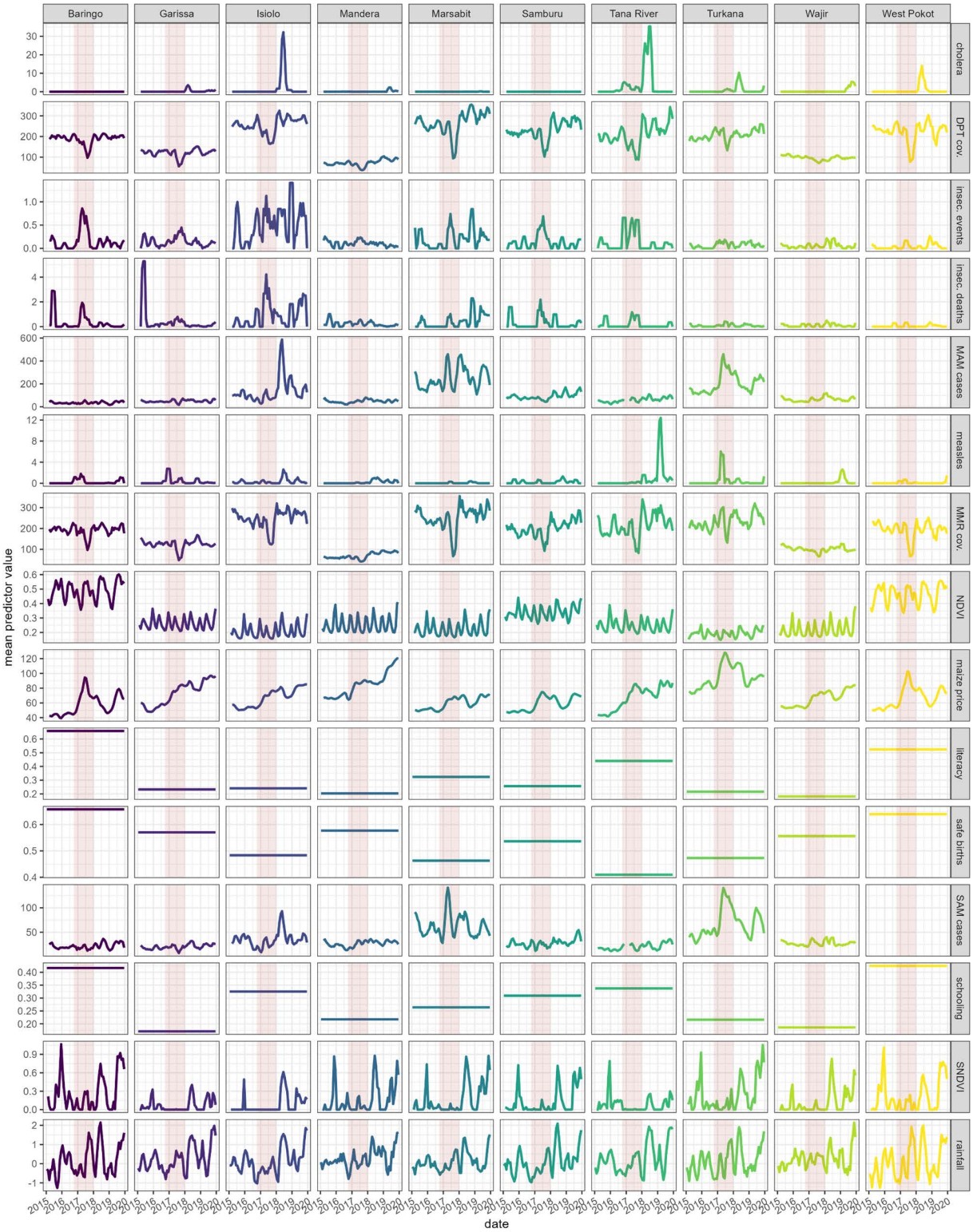

**Fig 4. Trends in each predictor variable, by county.** The pink-shaded band indicates the drought period (October 2016 to December 2018). DPT = pentavalent vaccine. MAM = moderate acute malnutrition. MMR = measles, mumps, rubella vaccine. **(S)**NDVI = (Standardised) Normalised Difference Vegetation Index. SAM = severe acute malnutrition. Cov. = coverage. Insec. = insecurity. See Table C in S1 Appendix for predictor units.

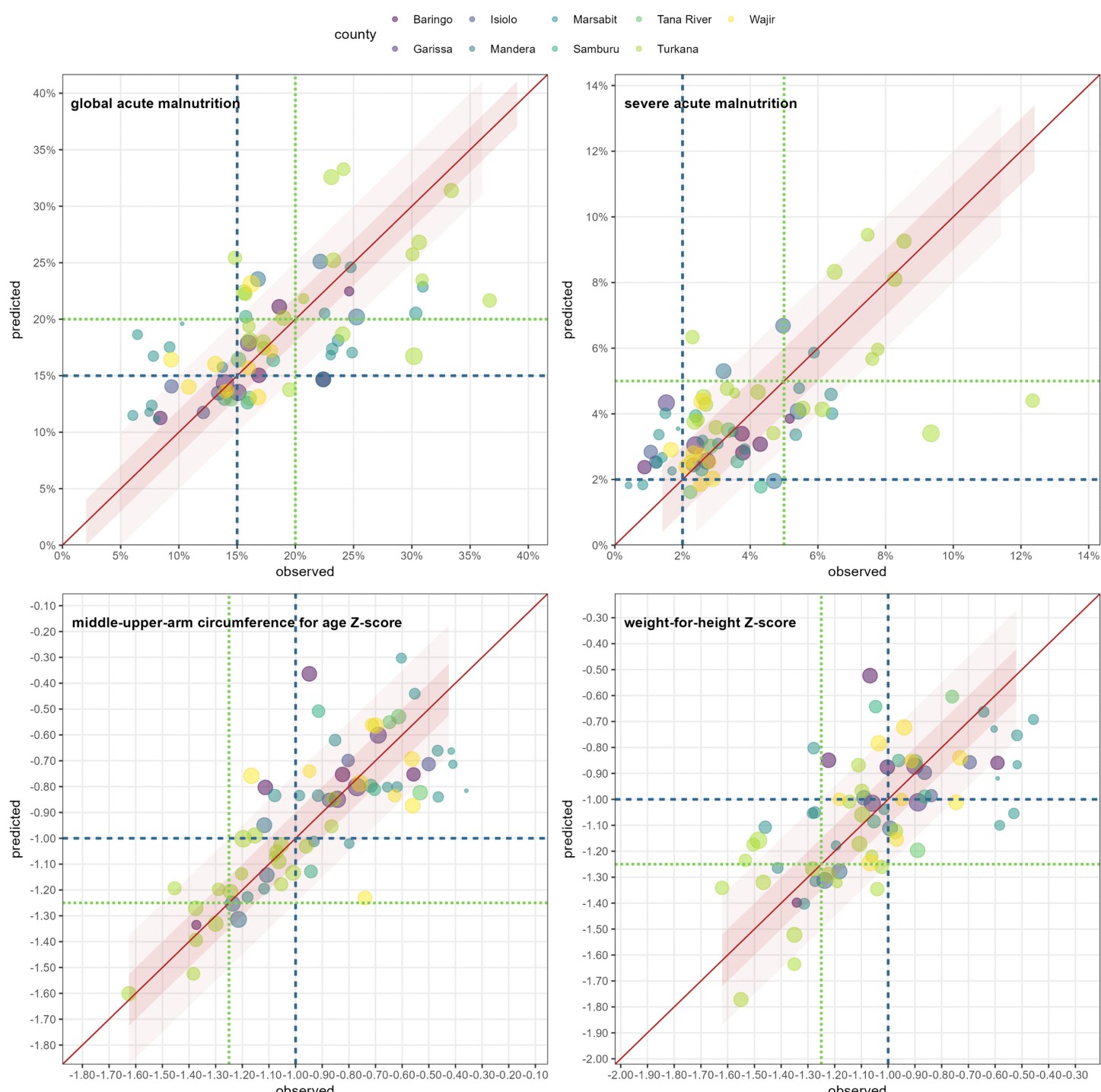

**Fig 5. Generalised linear or additive model predictions versus observations for each fold in leave-one-out cross-validation, by outcome.**
Within each graph, dots represent individual survey strata left out of the training sample and which the model was validated on. The size of each dot is proportional to the number of child-observations, and the colour maps to the county. The diagonal line denotes perfect fit, while shaded areas above and beyond it show alternative error thresholds. Finally, horizontal and vertical lines denote interesting threshold values of the outcome for which the model's sensitivity was computed.

**Table 2. Summary statistics of generalised linear or additive model performance, by outcome.**

| characteristic | GAM prevalence[a] | | SAM prevalence[b] | | mean WHZ[c] | | mean MUACZ[c] | |
|---|---|---|---|---|---|---|---|---|
| Mean absolute bias | +0.20% | | 0.10% | | -0.01 | | -0.01 | |
| Mean absolute error | 4.50% | | 1.40% | | 0.19 | | 0.15 | |
| Proportion of predictions within a given error bound of observations‡ | ±2% | 32.4% (22/68) | ±1% | 39.7% (27/68) | ±0.10 | 29.4% (20/68) | ±0.10 | 44.1% (30/68) |
| | ±5% | 60.3% (41/68) | ±2% | 85.3% (58/68) | ±0.25 | 72.1% (49/68) | ±0.25 | 82.4% (56/68) |
| Sensitivity for detection of threshold values‡ | ≥15% | 84.8% (39/46) | ≥2% | 92.7% (51/55) | <-1 | 78.0% (32/41) | <-1 | 81.5% (22/27) |
| | ≥20% | 66.7% (16/24) | ≥5% | 41.2% (7/17) | <-1.25 | 58.8% (10/17) | <-1.25 | 75.0% (6/8) |

Predictors:

[a]cholera incidence rate (previous 3mths rolling mean), insecurity event rate (3mths), MAM admissions rate (3mths), measles-mumps-rubella dose 1 vaccination rate (6mths), SNDVI (6mths), proportion of the population with access to safe assisted births, standardized precipitation index (6mths), white maize price (3mths).

[b]as for GAM prevalence, but SAM admissions rate used instead of MAM admissions rate.

[c]as for GAM prevalence, but with additive thin-plate smoothing terms for measles-mumps-rubella dose 1 vaccination rate, SNDVI, standardized precipitation index (6mths) and white maize price (3mths).

‡Numbers in parentheses are the number of correct predictions out of all observations. While 79 possible LOOCV folds (survey strata) were available, 11 did not contain all the predictor variables or levels used to train random forest models, leaving a denominator of observations of 68.

droughts. Our study explored a predictive modelling approach to complement ground surveillance. None of the models evaluated, regardless of outcome (nutritional indicator), offered compellingly high performance. While sensitivity was reasonably high for the lower of two thresholds against which it was benchmarked, this threshold included a majority of observed point estimates, i.e., may not be of great utility in the Kenyan context to detect a marked deterioration from baseline. Only the random forest model for GAM prevalence offered reasonably high sensitivity for the higher threshold of ≥20% prevalence, which, at least within this pool of survey data, would indeed identify unusually elevated malnutrition.

### Findings in context

While largely disappointing, the performance of models in Kenya, albeit underpinned by different predictor variables, is higher than in Somalia and South Sudan based on a previous study [18]. Other attempts at predicting childhood malnutrition at subregional resolution have had mixed success. A model of stunting based on demographic and health survey data and remotely sensed predictors performed reasonably well in Bangladesh but not Ghana [36]. A landmark small-area estimation study [37] of malnutrition across Africa, relying on large-scale countrywide surveys and a wide range of static and time-varying predictors, achieved high performance for stunting and reasonable performance for wasting, although the predictors and data included in this study may not offer the kind of temporal resolution required to predict deteriorations in nutritional status within the relatively short timeframe of a developing drought. In Bangladesh, random forest models outperformed other approaches for predicting maternal undernutrition, but these are based on individual-level predictors collected during demographic and health surveys, i.e., do not circumvent the need for data collection and may be more suitable for individual screening and case identification [38].

### Limitations

This analysis is mainly limited by the quantity and quality of available data that could be used to develop and evaluate models. While SMART data had good quality, the time period investigated encompassed a single drought period, and in some counties even the drought was not correlated with dramatic changes in nutritional indicators, leaving relatively limited statistical variability with which to discriminate between alternative models. The analysis also relied on a limited

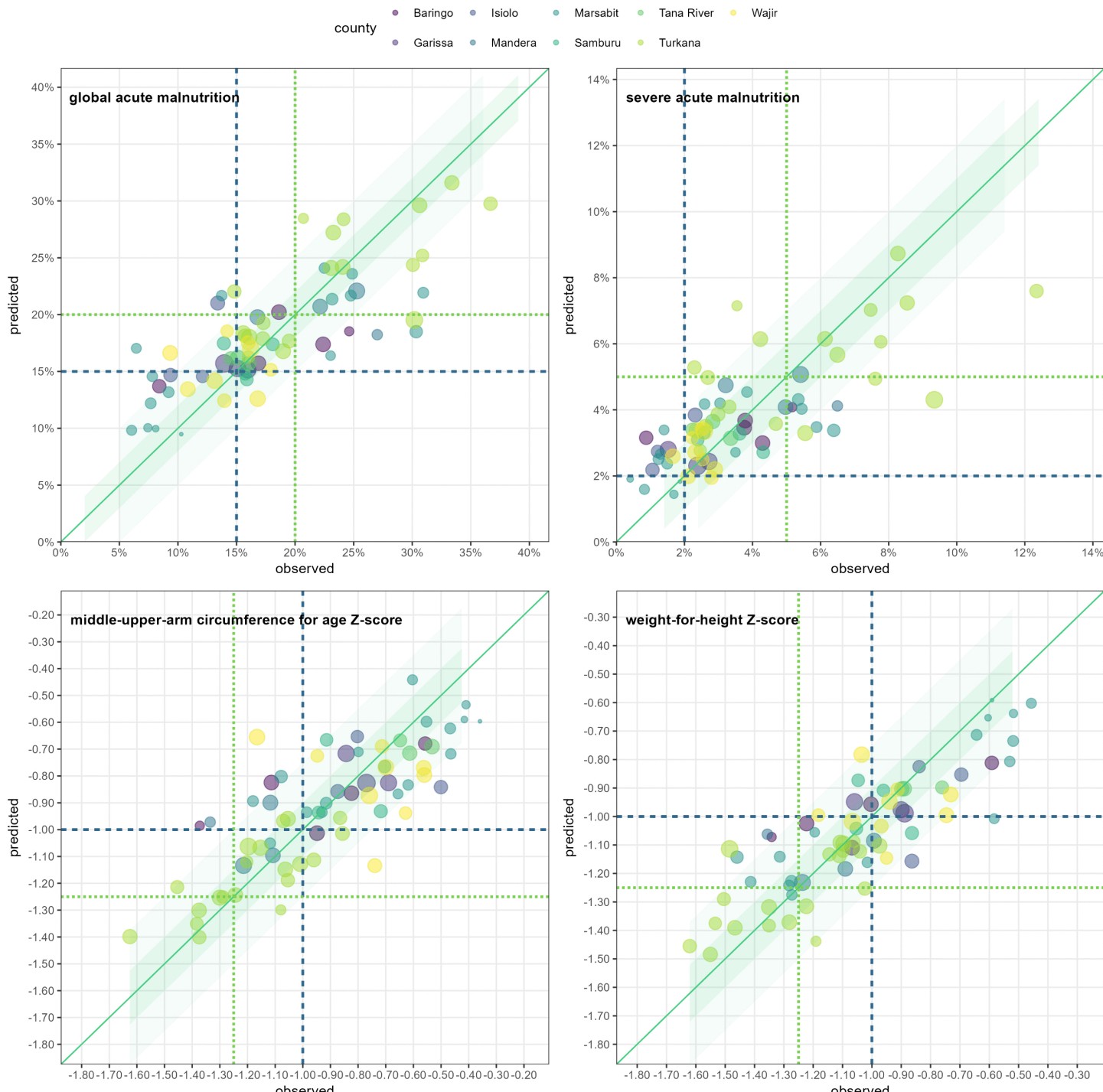

**Fig 6. Random forest predictions versus observations for each fold in leave-one-out cross-validation, by outcome.** Forests were grown by splitting data by up to **three** variables at each tree node. Within each graph, dots represent individual survey strata left out of the training sample and which the model was validated on. The size of each dot is proportional to the number of child-observations, and the colour maps to the county. The diagonal line denotes perfect fit, while shaded areas above and beyond it show alternative error thresholds. Finally, horizontal and vertical lines denote interesting threshold values of the outcome for which the model's sensitivity was computed.

**Table 3. Summary statistics of random forest model performance, by outcome.**

| characteristic† | GAM prevalence | | SAM prevalence | | mean WHZ | | mean MUACZ | |
|---|---|---|---|---|---|---|---|---|
| Mean absolute bias | +0.3% | | 0.0% | | -0.01 | | 0.00 | |
| Mean absolute error | 3.6% | | 1.2% | | 0.12 | | 0.15 | |
| Proportion of predictions within a given error bound of observations‡ | ±2% | 41.8% (28/67) | ±1% | 50.7% (34/67) | ±0.10 | 53.7% (36/67) | ±0.10 | 41.8% (28/67) |
| | ±5% | 71.6% (48/67) | ±2% | 82.1% (55/67) | ±0.25 | 86.6% (58/67) | ±0.25 | 86.6% (58/67) |
| Sensitivity for detection of threshold values‡ | ≥15% | 91.1% (41/45) | ≥2% | 96.3% (52/54) | <-1 | 87.5% (35/40) | <-1 | 66.7% (18/27) |
| | ≥20% | 73.9% (17/23) | ≥5% | 47.1% (8/17) | <-1.25 | 52.9% (9/17) | <-1.25 | 44.4% (4/9) |

†All models feature the following predictors: cholera incidence rate (previous 3mths rolling mean), insecurity event rate (3mths), MAM admission rate (3mths), measles incidence rate (3mths), measles-mumps-rubella dose 1 vaccination rate (6mths), NDVI (6mths), pentavalent dose 3 vaccination rate (6mths), proportion of children enrolled in school, proportion of literate reproductive-age women, proportion of the population with access to safe assisted births, standardized precipitation index (6mths), white maize price (3mths).

‡Numbers in parentheses are the number of correct predictions out of all observations. While 79 possible LOOCV folds (survey strata) were available, 12 did not contain all the predictor variables or levels used to train random forest models, leaving a denominator of observations of 67.

range of predictors, reflecting datasets found to have consistent availability and geographic specification over the region and period analysed. It is plausible that additional data on likely correlates of malnutrition, including endemic morbidity, and more specific and granular data on food insecurity, e.g., terms of trade (purchasing power) or adoption of household coping mechanisms, would have improved predictive performance. These indicators were not available at the sub-county level with sufficient temporal granularity to permit linkage with SMART survey strata and capture the fluctuations caused by relative drought periods over the time period analysed, as evidenced by the patterns in available predictors (Fig 4). We therefore opted to include only those indicators that were both plausibly related to malnutrition and consistently available across space and time, to support operational use in future predictive modelling applications. In some cases, where key indicators were conceptually relevant but not consistently available from administrative sources (e.g., DHIS2), we relied on publicly available modelled proxies to preserve coverage and alignment with SMART survey timing. For example, WorldPop-derived estimates of access to safe assisted births were used as a proxy for maternal health service availability, given the lack of reliable sub-county data from health information systems. While these proxies allowed broader spatial modelling and reproducibility, we acknowledge that they may only indirectly capture the intended constructs and could introduce additional measurement error. Their inclusion was a pragmatic compromise to enable scalable, operationally useful prediction.

Generally, any error in both anthropometric data (e.g., due to inadequate measurement practices) and predictors would have affected model performance, most probably by 'diluting' regression coefficients and thus under-estimating the true predictive contribution of each predictor.

Northern Kenya has since experienced a further drought (2022–2023), and it would have been useful to supplement the dataset with observations from this more recent crisis, or to test the model's performance in predicting SAM and GAM prevalence against SMART survey results. However, our project was funded and had data access permissions only for the 2016–2018 drought and preceding years.

Apart from the data available, the study did not investigate the performance of alternative models that may be more suitable for prediction with many co-variates (e.g., lasso regression, which addresses instances of extensive predictor correlation; note however that we did not observe serious correlation among most predictors; see Fig D in S1 Appendix) or machine learning approaches such as neural networks, though the latter, like random forests, would likely encounter challenges predicting out of sample, and may thus require careful fine-tuning. These approaches might have been superior to building models based on predictors' AIC ranking. The relationships between predictors and the outcomes investigated may themselves not be constant in time, and it is therefore possible that a given model may perform well over the period

from which data to evaluate it were sourced, but less so later. Similarly, our findings should only be considered applicable to the Kenyan context, as predictor-outcome correlations may be differently modulated by other variables in other settings.

## Conclusions

We identified generalized linear and, perhaps more promisingly, machine learning (random forest) models that yielded moderate predictive performance for different indicators of acute childhood malnutrition in Kenya. The models for GAM in particular could potentially be used in lieu of educated guesses to support real-time situational awareness of and decision-making on deteriorating nutrition, but their superiority to experts (e.g., as estimated through structured expert elicitation) is unclear and any use of the models would need to be accompanied by training of decision-makers to ensure that they take into account model uncertainty when interpreting predictions.

It is plausible that these models could be improved if trained on a longer time series of data (critically, these should contain several peaks and troughs in ground-measured acute malnutrition burden, i.e., more variability in the outcomes studied, as would be the case if data were updated until the present time, thereby also capturing the 2022–2023 drought); and if a wider range of consistently available predictor data were harnessed, including more indicators of morbidity and health service performance, as well as measures of food insecurity other than market prices. A predictive modelling approach remains attractive in contexts, such as northern Kenya, where drought is a known threat and the costs and feasibility of ongoing primary data collection across the affected region are a serious constraint to surveillance.

## Supporting information

**S1 Appendix. Supplementary tables and figures.**
(DOCX)

## Acknowledgments

We are grateful to UNICEF and the Ministry of Health of Kenya for sharing non-public datasets used for this analysis. We also extend our appreciation to Séverine Frison for her contributions to this work.

## Author contributions

**Conceptualization:** Francesco Checchi, Mara Nyawo, Lucy Maina.

**Data curation:** Francesco Checchi, Rahaf AbuKoura, Mara Nyawo, Lucy Maina.

**Formal analysis:** Francesco Checchi.

**Funding acquisition:** Francesco Checchi, Suneetha Kadiyala.

**Investigation:** Francesco Checchi.

**Methodology:** Francesco Checchi.

**Project administration:** Francesco Checchi.

**Writing – original draft:** Francesco Checchi.

**Writing – review & editing:** Rahaf AbuKoura, Suneetha Kadiyala, Mara Nyawo.

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
