## [Decision Letter · Decision Letter 0]

26 Jun 2025

PGPH-D-25-00663

Predicting the burden of acute malnutrition in drought-prone regions of Kenya: a statistical analysis

Dear Dr. Checchi,

Thank you for submitting your manuscript to PLOS Global Public Health. After careful consideration, we feel that it has merit but does not fully meet PLOS Global Public Health’s publication criteria as it currently stands. Therefore, we invite you to submit a revised version of the manuscript that addresses the points raised during the review process.

Please note that we have only been able to secure a single reviewer to assess your manuscript. We are issuing a decision on your manuscript at this point to prevent further delays in the evaluation of your manuscript. Please be aware that the editor who handles your revised manuscript might find it necessary to invite additional reviewers to assess this work once the revised manuscript is submitted. However, we will aim to proceed on the basis of this single review if possible.

We look forward to receiving your revised manuscript.

Kind regards,

Jianhong Zhou

Staff Editor

Journal Requirements:

1. Please insert an Ethics Statement at the beginning of your Methods section, under a subheading 'Ethics Statement'.

i. State the initials, alongside each funding source, of each author to receive each grant.

ii. State what role the funders took in the study. If the funders had no role in your study, please state: “The funders had no role in study design, data collection and analysis, decision to publish, or preparation of the manuscript.”

3. Some material included in your submission may be copyrighted. According to PLOS’s copyright policy, authors who use figures or other material (e.g., graphics, clipart, maps) from another author or copyright holder must demonstrate or obtain permission to publish this material under the Creative Commons Attribution 4.0 International (CC BY 4.0) License used by PLOS journals. Please closely review the details of PLOS’s copyright requirements here: PLOS Licenses and Copyright. If you need to request permissions from a copyright holder, you may use PLOS's Copyright Content Permission form.

Potential Copyright Issues:

Figure 1: please (a) provide a direct link to the base layer of the map (i.e., the country or region border shape) and ensure this is also included in the figure legend; and (b) provide a link to the terms of use / license information for the base layer image or shapefile. We cannot publish proprietary or copyrighted maps (e.g. Google Maps, Mapquest) and the terms of use for your map base layer must be compatible with our CC-BY 4.0 license.

Additional Editor Comments (if provided):

Reviewers' comments:

Reviewer's Responses to Questions

**Comments to the Author**

1. Does this manuscript meet PLOS Global Public Health’s publication criteria? Is the manuscript technically sound, and do the data support the conclusions? The manuscript must describe methodologically and ethically rigorous research with conclusions that are appropriately drawn based on the data presented.? Is the manuscript technically sound, and do the data support the conclusions? The manuscript must describe methodologically and ethically rigorous research with conclusions that are appropriately drawn based on the data presented.

Reviewer #1: Yes

2. Has the statistical analysis been performed appropriately and rigorously?

Reviewer #1: Yes

3. Have the authors made all data underlying the findings in their manuscript fully available (please refer to the Data Availability Statement at the start of the manuscript PDF file)?

The PLOS Data policy requires authors to make all data underlying the findings described in their manuscript fully available without restriction, with rare exception. The data should be provided as part of the manuscript or its supporting information, or deposited to a public repository. For example, in addition to summary statistics, the data points behind means, medians and variance measures should be available. If there are restrictions on publicly sharing data—e.g. participant privacy or use of data from a third party—those must be specified.requires authors to make all data underlying the findings described in their manuscript fully available without restriction, with rare exception. The data should be provided as part of the manuscript or its supporting information, or deposited to a public repository. For example, in addition to summary statistics, the data points behind means, medians and variance measures should be available. If there are restrictions on publicly sharing data—e.g. participant privacy or use of data from a third party—those must be specified.

Reviewer #1: Yes

4. Is the manuscript presented in an intelligible fashion and written in standard English?

Reviewer #1: Yes

Reviewer #1: The authors may take note of the following comments and provide relevant reasons and background for not including critical indicators based on the causality framework that they are referring to in their references. While their previous study based on similar work done in South Sudan and Somalia, published in BMC Nutr. in 2022 may be relevant in those countries as they are relatively data-poor, a similar or somewhat identical approach in a data rich country such as Kenya, does not entirely make sense.

I have specific comments related tot he section on Predictors of Malnutrition that the authors may wish to clarify. Further, i have some minor comment related to the SMART survey costs that they note based on global estimates.

Line 62-63 - the authors quote a cost for SMART survey as being $10-40,000. However, they themselves have done the analysis of sub-county level SMART surveys. It may be better to compute the actual costs of the SMART surveys that they have actually used instead of stating global national SMART survey costs.

Section on Predictors of Malnutrition

In Line 112, authors refer to causal frameworks in previous studies (References 18 and 22); While the quoted reference 18 and the figures do not refer to the causal frameworks, the reference no. 12 does have a causal framework in page 11 (Figure 2). Critical proximate determinants of acute malnutrition include - inadequate dieteray intake and disease (immediate casues), household food insecurity, inadequate care, unhealthy household environment and lack of health services as critical proximate causes. Several of the items in the Table 1 as predictors do not address these critical set of indicators. For example, food security measured through price of food staples is not same as household food insecurity. The study did not include any indicator either direct or proxy on either care or unhealthy household environment. If they had made an attempt, it was possible to get information on some of these from hospitals and health centres in the sub-county level, as well as other surveys possibly at the county level (including e.g. the county level data sheets that are available for DHS-Kenya). The list of predictors in Table 1 are predominantly distal or remote macro-level indicators at least from the perspective of the conceptual framework for acute malnutrition. The authors note that their analysis is in some ways a repetition of what they found elsewhere as published by most of the same authors in South Sudan and Somalia. Compared to those countries, Kenya counties have lot more data from both DHIS2 and other sources on several of the direct or proxy indicators for immediate and underlying causes instead of depending on predominantly basic or remote/macro level indicators as they possibly had to in data poor countries such as Somalia and South Sudan. Authors may wish to clarify why they did not seek to include direct or relevant indicators that can explain the causality better?

One should have looked at data related to caring practices, household water and sanitation access, handwashing from closer to the community sources and previous county level reports as well as HMIS data. How would data on for example - proportion of population with access to safe assisted births from "WorldPop" be more valuable than data and information that is available from the counties own records? A review of the table 1 suggests that data from global sources on indicators which at best can only marginally influence malnutrition outcomes have been given importance somewhat more than other possible indicators which have not even been considered. The authors may wish to clarify the reason for such exclusion of critical indicators and for inclusion of indicators which may have some marginal influence only.

Rest of the paper heavily depends on the clarification that the authors will need to provide, before i can sufficiently comment on the detailed analysis and presenatation as well as on the conclusions of the study based on indicators which do not necessarily fit into the causality framework adequately.

**Do you want your identity to be public for this peer review?** For information about this choice, including consent withdrawal, please see our Privacy Policy..

Reviewer #1: **Yes:**Lakshmi Narasimhan BalajiLakshmi Narasimhan BalajiLakshmi Narasimhan BalajiLakshmi Narasimhan Balaji

---

## [Decision Letter · Decision Letter 1]

8 Oct 2025

PGPH-D-25-00663R1

Predicting the burden of acute malnutrition in drought-prone regions of Kenya: a statistical analysis

Dear Dr. Checchi,

Thank you for submitting your manuscript to PLOS Global Public Health. After careful consideration, we feel that it has merit but does not fully meet PLOS Global Public Health’s publication criteria as it currently stands. Therefore, we invite you to submit a revised version of the manuscript that addresses the points raised during the review process.

I sought the opinion of a statistician reviewer to contribute more insights to the previous reviewer. This means that the manuscript has now been reviewed by 2 independent reviewers, as required. I agree with the new reviewer that these aspects below need to be addressed first before full acceptance of the manuscript for publication.

We look forward to receiving your revised manuscript.

Kind regards,

Gerard Bryan Gonzales

Academic Editor

Journal Requirements:

Additional Editor Comments (if provided):

Reviewers' comments:

Reviewer's Responses to Questions

**Comments to the Author**

Reviewer #2: All comments have been addressed

publication criteria? Is the manuscript technically sound, and do the data support the conclusions? The manuscript must describe methodologically and ethically rigorous research with conclusions that are appropriately drawn based on the data presented.? Is the manuscript technically sound, and do the data support the conclusions? The manuscript must describe methodologically and ethically rigorous research with conclusions that are appropriately drawn based on the data presented.

Reviewer #2: Yes

3. Has the statistical analysis been performed appropriately and rigorously?

Reviewer #2: Yes

4. Have the authors made all data underlying the findings in their manuscript fully available (please refer to the Data Availability Statement at the start of the manuscript PDF file)?

The PLOS Data policy requires authors to make all data underlying the findings described in their manuscript fully available without restriction, with rare exception. The data should be provided as part of the manuscript or its supporting information, or deposited to a public repository. For example, in addition to summary statistics, the data points behind means, medians and variance measures should be available. If there are restrictions on publicly sharing data—e.g. participant privacy or use of data from a third party—those must be specified.requires authors to make all data underlying the findings described in their manuscript fully available without restriction, with rare exception. The data should be provided as part of the manuscript or its supporting information, or deposited to a public repository. For example, in addition to summary statistics, the data points behind means, medians and variance measures should be available. If there are restrictions on publicly sharing data—e.g. participant privacy or use of data from a third party—those must be specified.

Reviewer #2: Yes

5. Is the manuscript presented in an intelligible fashion and written in standard English?

Reviewer #2: Yes

Reviewer #2: -What was the age range of the children included?

-The surveys were conducted between January 2015 to December 2019, however, no details have been provided on number of surveys included and the exact dates/month of the year when data were collected. Acute undernutrition is very seasonal, so is drought and rainfall in Kenya/tropical. Having a table showing the list of the surveys, dates and drought periods would be very informative.

-While using AIC for initial variable screening is reasonable, relying solely on univariate AIC risks excluding variables that may be weak alone but strong in combination (due to interactions or confounding).

-Consider discussing this limitation or complementing AIC screening with other selection methods (e.g., lasso, stepwise with interactions, or expert-driven selection).

-The text mentions "rolling-mean periods" for predictors, but doesn't clarify the time window used (e.g., 1-month, 3-month rolling averages?) or how lags were tested. This is especially critical in modeling nutrition as predictor-outcome relationships are often delayed (e.g., rainfall → harvest → malnutrition).

-The stepwise addition of variables based on AIC is common but susceptible to overfitting, especially without regularization. Was any constraint placed on the maximum number of predictors to avoid overly complex models?

-For the random forests: using only 3 variables per split node is quite low — why was this chosen? Did you test performance with different values?

-Were assumptions of the GLMs/GAMs assessed (e.g., residual distribution, homoscedasticity for continuous outcomes)? Some brief mention would strengthen confidence in model validity.

-It’s unclear how “alternative crisis severity thresholds” were defined (e.g., standard IPC thresholds for GAM >10%, 15%?) lines 171/172. Including these definitions and rationales would aid interpretability.

-Were the random forest models formally compared with GLMs model? Or how was `marginally better’ line 254 comments based on?

-The authors notes poor performance of the models. Any thoughts on why the model poorly performed and alternatives would greatly improve the model.

**Do you want your identity to be public for this peer review?** For information about this choice, including consent withdrawal, please see our Privacy Policy..

Reviewer #2: **Yes:**Dr Moses NgariDr Moses NgariDr Moses NgariDr Moses Ngari

---

## [Decision Letter · Decision Letter 2]

23 Feb 2026

PGPH-D-25-00663R2

Predicting the burden of acute malnutrition in drought-prone regions of Kenya: a statistical analysis

Dear Dr. Checchi,

Thank you for submitting your manuscript to PLOS Global Public Health. After careful consideration, we feel that it has merit but does not fully meet PLOS Global Public Health’s publication criteria as it currently stands. Therefore, we invite you to submit a revised version of the manuscript that addresses the points raised during the review process.

The reviewers have suggested some further minor revisions in their comments below. Specifically they have requested additional clarifying sentences to help the readers' understanding and also suggested making the text more concise where possible. Please review their comments and make the appropriate revisions.

We look forward to receiving your revised manuscript.

Kind regards,

Emma Campbell, Ph.D

Staff Editor

Journal Requirements:

Additional Editor Comments (if provided):

Reviewers' comments:

Reviewer's Responses to Questions

**Comments to the Author**

Reviewer #3: All comments have been addressed

Reviewer #4: (No Response)

publication criteria? Is the manuscript technically sound, and do the data support the conclusions? The manuscript must describe methodologically and ethically rigorous research with conclusions that are appropriately drawn based on the data presented.? Is the manuscript technically sound, and do the data support the conclusions? The manuscript must describe methodologically and ethically rigorous research with conclusions that are appropriately drawn based on the data presented.

Reviewer #3: Yes

Reviewer #4: Yes

3. Has the statistical analysis been performed appropriately and rigorously?

Reviewer #3: Yes

Reviewer #4: Yes

4. Have the authors made all data underlying the findings in their manuscript fully available (please refer to the Data Availability Statement at the start of the manuscript PDF file)?

The PLOS Data policy requires authors to make all data underlying the findings described in their manuscript fully available without restriction, with rare exception. The data should be provided as part of the manuscript or its supporting information, or deposited to a public repository. For example, in addition to summary statistics, the data points behind means, medians and variance measures should be available. If there are restrictions on publicly sharing data—e.g. participant privacy or use of data from a third party—those must be specified.requires authors to make all data underlying the findings described in their manuscript fully available without restriction, with rare exception. The data should be provided as part of the manuscript or its supporting information, or deposited to a public repository. For example, in addition to summary statistics, the data points behind means, medians and variance measures should be available. If there are restrictions on publicly sharing data—e.g. participant privacy or use of data from a third party—those must be specified.

Reviewer #3: Yes

Reviewer #4: Yes

5. Is the manuscript presented in an intelligible fashion and written in standard English?

Reviewer #3: Yes

Reviewer #4: Yes

Reviewer #3: This is a strong and well-thought study that makes a valuable contribution to the literature on nutrition surveillance in drought-affected countries. The cautious interpretation of findings and emphasis on operational relevance are appropriate. Minor revisions could further strengthen the manuscript, including brief clarification of how the models might be used in practice for early warning or situational awareness and light editing for conciseness. Overall, the manuscript is suitable for publication after minor revisions.

Reviewer #4: This study evaluates predictive approaches, namely generalized linear and additive models as well as random forest methods, to estimate acute malnutrition using routinely collected and/or publicly available data in combination with SMART survey data in Kenya. The analysis covers 2015–2019, a period that included one major drought episode. The authors examine global acute malnutrition (GAM), severe acute malnutrition (SAM), middle-upper-arm circumference-for-age Z-score, and weight-for-height Z-score as outcomes. Predictors were selected using AIC criteria, and model performance was evaluated using cross-validation. Overall, predictive performance was moderate, with the most promising results obtained from the random forest model for GAM.

The rationale of the study is highly compelling. The possibility of using routinely collected or publicly available data in drought-prone settings to estimate malnutrition burden, rather than relying on educated guesses or costly field surveys, is of clear operational importance. The manuscript is part of a series of studies attempting to predict malnutrition burden, and in this Kenyan setting the authors had access to more extensive and granular data than in previous applications. Despite this, predictive accuracy remains modest, which in itself is an important and informative finding.

In my view, the study is highly relevant. The authors have addressed previous reviewer comments carefully and have added clarifications that substantially improve transparency, particularly regarding predictor selection, proxy measures, and trade-offs between conceptual relevance and operational feasibility. The discussion of limitations is candid and balanced. The manuscript could still be made slightly more accessible for readers less familiar with predictive modelling, for example by briefly clarifying the rationale for AIC-based model selection and the interpretation of cross-validation results. However, the current version already does a solid job in guiding the reader through the analytical strategy.

The strongest limitation, as acknowledged by the authors, is that the study period includes only one major drought. This necessarily restricts variability in the outcome and limits the ability to assess whether predictor–outcome relationships would remain stable across multiple drought cycles. In this light, it would be helpful if the authors could clarify why more recent data, particularly from the 2022–2023 drought, were not incorporated. Even if data availability or compatibility constraints prevented this, a short explanation would strengthen the manuscript.

With respect to modelling strategy, the authors rely on AIC-based selection for generalized models and compare these with random forest approaches. This is a reasonable and transparent strategy, particularly given the modest sample size and the operational nature of the research question. One additional sentence elaborating on why penalized regression approaches such as lasso were not pursued could be helpful, especially since such methods are often considered when handling multiple correlated predictors. That said, the authors’ argument that alternative machine learning methods would likely face similar structural constraints, given the limited temporal variability and measurement error in predictors, is plausible and does not detract from the core findings.

The cross-validation approach at the survey-stratum level is appropriate and clearly described. Readers may benefit from a brief reflection in the discussion on how predictive performance might differ under alternative validation schemes, for example temporal holdout designs, although this is not essential for publication.

Overall, this study makes a meaningful contribution to the literature on predictive modelling of malnutrition burden. Importantly, it demonstrates both the potential and the current limits of such approaches when constrained to routinely available data in real-world humanitarian settings. The manuscript does not overstate its findings and appropriately frames them as part of an ongoing effort to improve predictive tools. I believe the study is suitable for publication and will be of interest to researchers and practitioners working at the intersection of nutrition surveillance, humanitarian response, and predictive analytics.

**Do you want your identity to be public for this peer review?** For information about this choice, including consent withdrawal, please see our Privacy Policy..

Reviewer #3: No

Reviewer #4: No

---

## [Editor Report · Decision Letter 3]

12 Mar 2026

Predicting the burden of acute malnutrition in drought-prone regions of Kenya: a statistical analysis

PGPH-D-25-00663R3

Dear Dr Checchi,

We are pleased to inform you that your manuscript 'Predicting the burden of acute malnutrition in drought-prone regions of Kenya: a statistical analysis' has been provisionally accepted for publication in PLOS Global Public Health.

Best regards,

Julia Robinson

Executive Editor